# The Bugs in the Bags: The Risk Associated with the Introduction of Small Quantities of Fruit and Plants by Airline Passengers

**DOI:** 10.3390/insects13070617

**Published:** 2022-07-10

**Authors:** Roberta Pace, Roberta Ascolese, Fortuna Miele, Elia Russo, Raffaele V. Griffo, Umberto Bernardo, Francesco Nugnes

**Affiliations:** 1Institute for Sustainable Plant Protection, National Research Council (CNR), 80055 Portici, Italy; roberta.pace@ipsp.cnr.it (R.P.); roberta.ascolese@ipsp.cnr.it (R.A.); fortuna.miele@ipsp.cnr.it (F.M.); umberto.bernardo@ipsp.cnr.it (U.B.); 2Department of Agricultural Sciences, University of Naples “Federico II”, 80055 Portici, Italy; elia.russo@unina.it; 3Plant Health Service —Campania Region, 80124 Napoli, Italy; raffaele.griffo@regione.campania.it

**Keywords:** eggplant fruit and shoot borer, invasive alien species, monitoring, oriental fruit fly, passenger baggage, plant passport, quarantine pest, West Indian fruit fly

## Abstract

**Simple Summary:**

This study was carried out with the aim of emphasizing the importance of checking the plant material that can be imported in the baggage of airline passengers. Travelers are often unaware of the regulations in place and of the risks connected with such importation. The risk of the introduction of harmful organisms correlated with this pathway is yet not well studied and its frequency is underestimated. The results of the research underline the need for continuous checks at entry points and the establishment of a specialized position for inspections.

**Abstract:**

Among European countries, Italy is the most exposed to the risk of biological invasions, principally for its numerous entry points (ports and airports) and for climatic conditions favorable for the acclimatization of several invasive species. Here it was assessed that the greatest threats to our agro-ecosystems come mainly from the passenger baggage in which a variety of fruits and vegetables are carried. From 2016 to 2021, large quantities of plant products were found in the luggage of passengers travelling from outside the EU and seized at the BCPs (border control posts) in the Campania region. Inspections and the following laboratory analyses were conducted on the plant material to assess the presence of exotic pests. Inspections led to several non-native species being recorded, and among the intercepted organisms, some should be considered “alarming”, such as *Bactrocera dorsalis*, *Anastrepha obliqua*, and *Leucinodes africensis*. Despite a well-organized border inspection system, travelers transporting infested material unknowingly contribute to increasing the risk of the introduction of exotic species. Given the current situation, it is necessary to impose stricter controls and greater attention, ensuring compliance with the requirements of the new phytosanitary regulations by the actors involved in the transport of plant material. Finally, it is essential to improve awareness through a phytosanitary campaign on plant health risks, especially for people wishing to transport fruits and vegetables in their luggage.

## 1. Introduction

Worldwide trade and international travel are mainly responsible for the transfer of non-indigenous organisms among ecosystems [1,2,3]. These species are considered “invasive” when they can establish and cause economic and environmental damage in non-native areas [4]. Invader arthropod pests are perhaps the most pervasive component of global biodiversity loss and the destruction of ecosystem health [5,6,7,8]. In fact, some of these species are classified as “severe quarantine pests” because of their status in an area, widespread distribution, and invasive ability that can trigger serious trouble for agriculture worldwide, with consequent heavy economic losses [9]. It was estimated that invasive species introduced into the European Union (EU) cost about USD 13 billion per year [10].

Moreover, the number of alien arthropod species is predicted to increase sharply by 2050, particularly in European countries, due to the intensification of trade and international movement [11,12]. Indeed, sometimes the human impact on invasive processes is more dangerous than the international trade of green material and, most of the time, the threat is not adequately visible. In particular, invasive species are moved worldwide through human activities, and people are largely responsible for moving plant materials and their organisms from one country to another [2,13]. Relating the number of new introductions to activities such as trade, tourism, transport, and travel is a challenge. Calculating a precise estimation of the incidence of these activities on agriculture is not always possible because not all traffic is controlled, especially that which is untraceable [14]. The large amount of global movement, especially by air, makes it crucial to better understand how air passenger baggage represents a pathway for the introduction of new species into new territories [15]. Planning strategies to control these pests requires extensive characterization with some detailed information about biology, geographical distribution, and genetics [16].

With the purpose of improving reporting and facilitating an enhanced level of protection of the EU against harmful organism outbreaks, the European Commission has developed a web-based notification system, EUROPHYT [17]. In the recent years, many organisms were reported to EUROPHYT and successively classified as “new introduction”, “transient” or “present” in many territories of European states. Some species, after being classified as harmful, were listed in many databases of pests recommended for regulation as quarantine pests. Most of these organisms were introduced accidentally, while only 14% of introductions were intentional, with most of these for biological control programs [18].

Italy is one of the European countries most prone to the risk of the introduction of exotic species due to high migratory flows, its geographical position, and favorable climatic conditions for acclimatization [19]. More than 15 years ago, it was already estimated that more than 160 exotic pests have been introduced into Italy, and most of them were acclimatized, representing a real problem for ornamental and woody plants and, above all, for horticultural crops and fruit orchards [20]. These migratory flows of invasive species have increased in the last decade [21]. To date, Italy is still known as the European country with the most exotic taxa [22], and several of these have now spread widely throughout Italian territories.

The numerous entry points (ports and airports), known as border control points (BCPs), are a mean for introducing plant pests [10,23]. Italy has 33 BCPs, which is one of the highest numbers among EU countries in terms of BCPs relative to the size of the country [24].

Some of the most alarming plant pests and pathogens recorded in Italy arrived in Europe, as follows: *Xylella fastidiosa* Wells, first recorded in 2013 in the Apulia Region [25] and added to the EPPO A2 list [26]; *Aromia bungii* Falderman (Coleoptera: Cerambycidae), recorded in 2012 for the first time in Campania [27] and included in EPPO A2 list [26]; and, the most alarming species, belonging to the *Bactrocera* genus (Diptera: Tephritidae), *Bactrocera dorsalis* Hendel and *Bactrocera latifrons* Hendel, first recorded in Italy in 2018 and 2019, respectively [28,29], and inscribed in the EPPO A1 list [30]. Moreover, *A. bungii*, *X. fastidiosa*, *B. dorsalis*, and *B. zonata* are EU quarantine pests listed as “priority pests” in the Annex of the Commission Delegated Regulations (EU) 2019/1702 [31].

Increased knowledge about the pathways by which non-indigenous plant pests arrive at the borders could provide a framework for developing testable hypotheses about economic or ecological factors related to invasion success [32]. Carrying fruits, vegetables, and parts of plants in baggage is considered an important pathway for the arrival of new insect species, especially for exotic homopterans and flies [15,33].

To avoid the occurrence of invasive species within the European Union, the importing and handling of living plants and plant-based products are regulated by the EU Plant Health Directives. The actual EU Plant Health Regulation Framework increases preventive actions [31,34,35,36,37,38,39,40,41,42] against the introduction of new pests from third countries, summarized in Appendix A.

Furthermore, when traveling to the EU from a non-EU country, carrying a limited quantity of fruit and vegetables is allowed only for personal consumption, but all the security requirements need to be followed (Appendix A) and the competent authorities, in cooperation with operators responsible for the points of entry, must organize specific official controls that aim at identifying the non-compliant goods [40]. In this scenario, the Plant Health Service of the Campania Region implemented border inspections to support the phytosanitary inspectors with highly specialized personnel from the Institute for Sustainable Plant Protection of the National Research Council of Italy (CNR-IPSP), the University of Naples “Federico II”, and the Council for Agricultural Research and Economics (CREA) by establishing a Regional Phytosanitary Coordination Unit with the URCoFi project (for the strengthening of supervision activities and the control of pests) (http:// http://www.ipsp.cnr.it/en/urcofi-2/ accessed on 8 June 2020).

Within the project, the CNR-IPSP staff collaborated and supported phytosanitary agents in several monitoring activities for priority pests. This collaboration was extended to the phytosanitary controls on imported plants at BCPs in the Campania Region, which potentially represent a high phytosanitary risk, as corroborated by several previous surveys conducted at the entry points [43] and highlighting the importance of a specialized control and the need for establishing quarantine centers. In addition to the plant material that is regularly imported, many vegetables and fruits, along with other types of plant derivatives, are introduced into the Campania region in the baggage of passengers coming from third countries, unaware of the risks associated with the introduction of new harmful organisms. Hence, passengers and their luggage are on the list of the main categories that constitute a pathway for the introduction of invasive alien species [44]. Consequently, phytosanitary border inspections are a key element and are often the last barrier where these harmful pests associated with plant material can be intercepted [45].

This study reports the invasive species intercepted at the Campania Region BCPs, from 2016 to 2021. This work aims to:-comprehend the number and magnitude of insect species found both in plant materials transported through the baggage of airline passengers and in fruits and plants introduced by authorized importers;-estimate the risk represented by introduction pathways based on the quantities and frequencies of intercepted pests;-suggest the actions to be taken to avoid other introductions of threatening pests, based on this small experience.

## 2. Materials and Methods

### 2.1. Survey Sites

The activities of monitoring, controlling, and interception of potentially dangerous plant material originating from countries that do not belong to the EU were carried out from 2016 to 2021 at the BCPs of the Campania Region. Campanian BCPs are set at the ports of Naples and Salerno and at the International Airport of Capodichino (Naples). The presence of phytosanitary agents (PAs) at port BCPs was guaranteed daily, whereas the airport BCP was manned by PAs after the custom authority’s request due to the findings of copious quantities of infested plant material from countries with high phytosanitary risk.

Controls were divided into two categories: (i) plant material imported for trade at ports and the airport and (ii) plant material imported by airline passengers at the airport. All the procedures concerning the use of live organisms were performed at the containment facilities of the laboratory of IPSP.

### 2.2. Inspections of Plant Material Imported

#### 2.2.1. Plant Material Imported for Trade

Controls to regulated material [35,36] were carried out at the BCPs of the Campania Region in support of the Regional Phytosanitary Service and with the issue of the relative clearances.

Upon consignment arrival, the cargo was piped into a containment facility and the responsible border operator verified that the goods complied with phytosanitary standards, including those for shipment integrity (Figure 1). The subsequent checks were carried out assisted by URCoFi personnel, which ensured that the plant materials were in a good condition of health and free from pests. First, a careful visual inspection was carried out on the plant material to detect the presence or various symptoms and signs attributable to potentially harmful organisms (rottenness, ovipositional or trophic signs, immature stages, flying adult insects, dead insects).

All the control procedures followed the guidelines for phytosanitary control for import of the MIPAAF (Ministry of Agriculture, Food and Forestry Policies) (http://www.importfito.it/st_html/imp.html accessed on 6 September 2021) and the parameters predefined by the ISPM FAO-31 standard [46].

When potentially harmful organisms were found, samples were isolated in a storage box, hermetically sealed with a double envelope, and moved to IPSP laboratories to perform more in-depth analyses and morpho-molecular identification.

#### 2.2.2. Plant Materials Imported by Airline Passengers

Material from airline passengers containing vegetal materials (fruits, leaves, barks, woods, roots—transported for personal consumption or to local food shops) was seized by the customs authority due to the absence of the relevant phytosanitary certificate and was subsequently hermetically sealed in a double envelope and moved to the laboratories (Figure 1). When available, information on plant identity (common name or genus), quantity, provenience, and potentially useful information about the material were recorded.

### 2.3. Laboratory Activities

Collected vegetal materials were moved to the laboratory and identified following the available descriptions and botanical keys [47,48,49]. Performed analyses were:-detailed visual preliminary analysis to assess the signs of pest presence;-dissection of the material and its observation under a binocular lens to find any types of endophagous organisms;-identification through morphological and, when needed, molecular approaches.

At the end of all analyses, samples were destroyed.

Pests were identified following the available taxonomic keys, insect descriptions, and comparative images [50,51,52,53,54,55,56,57,58,59,60,61,62,63,64,65].

Morphological identification was not always possible, especially in the case of immature stages, due to the lack of distinct features. Hence, insects were kept in “bug dorm” (30 × 30 × 30 cm; BioQuip products, Rancho Dominguez, CA, USA) cages under controlled climatic conditions (25 ± 1 C°; UR 60 ± 10%) to await the suitable stage for identification (Figure 1).

When needed, the preliminary morphological identification was then integrated with the molecular analysis. Thus, useful molecular markers were chosen based on the literature concerning molecular identification at the family or genus level. Total DNA was isolated from each specimen using Chelex 100 (Bio-Rad, Richmond, CA, USA) and Proteinase K-based methods [66], and the component volumes were varied based on the sample size [67]. A portion of the *cytochrome c oxidase* subunit I (COI) and, when necessary, the nuclear gene ITS1 region were amplified using primer pairs and thermocycler conditions, as in [28].

PCR products were visualized on a 1.2% agarose gel stained with GelRED^®^ (Biotium, Fremont, CA, USA) or Xpert Green^®^ (GRiSP, Porto, Portugal) and directly sequenced.

Electropherograms were assembled with BIOEDIT 7.2.5 [68], individually checked “by eye” for ambiguous nucleotides, and virtually translated into amino acids to detect nonsense codons or frameshift mutations using EMBOSS Transeq (http://www.ebi.ac.uk/Tools/st/emboss_transeq/ accessed on 23 November 2021). The obtained sequences were blasted against homologous sequences available in GenBank (https://www.ncbi.nlm.nih.gov/genbank/ accessed on 3 December 2021) and the BOLD database and also through the BOLD Identification System (www.boldsystems. org accessed on 3 December 2021) to find similarities that allowed for proceeding with taxonomic identification. COI sequences were deposited in GenBank with accession numbers from OL693238 to OL693251 and from OL703028 to OL703031.

## 3. Results

### 3.1. Origins and Plant Materials

A total of 85 inspections were carried out at the BCPs in the Campania region between 2016 and 2021 by the specialist staff of the CNR-IPSP supporting the ordinary controls by phytosanitary service authorities (Table 1). Plant materials were largely seized at the BCP of the airport, although an amount of the material seized at both ports and the airport resulted in positive identification of infestations.

Plant materials came from three continents: Africa, Asia, and South America, with a distinct preponderance from Asia (78%). Among the African countries, Ghana, Morocco, and Nigeria stand out with percentages close to 20%, while the highest South American percentage was reached by Colombia. Above all, the large amount of confiscated material came from Bangladesh (~71% of the total from Asian countries) (Figure 2).

Seized material usually consisted of different quantities of fruits and different plant species (Appendix A). In fact, the incoming plant materials included especially thermophilic and mesothermal species, and, among the botanical families, Rutaceae, Myrtaceae, and Solanaceae were the most represented, with respective percentages of 13.26%, 9.32%, and 8.24% (Appendix A).

Considering there were also 10 positive inspections of unidentified material of plant origin (barks, dried leaves, ground spices), the inspections were positive for the presence of at least one arthropod in up to 40.7% of the total activities (Table 1, Figure 3).

About 50% of the incoming goods in which harmful pests were detected came from Asian countries. Among these, Bangladesh had the highest number of infested plant materials (47%). Percentages higher than 20% were reached by Ethiopia and Ghana among the African countries and by Argentina, Uruguay, and El Salvador in the South American countries (Figure 4).

Among the organisms, carpophagous and phytophagous (sapsucker) insects were the most frequently detected, followed by xylophagous insects (usually longhorns and ambrosia beetles) (Table 2). Hemiptera was found to be the most common insect order in the monitored material. Other arthropods such as Arachnida and Diplopoda, not of concerning phytosanitary interest, were also found (Figure 5 and Table 2). The plant material from Asia revealed the greatest parasite diversity, understood as the number of detected orders, except for Coleoptera and Orthoptera, which were mostly or solely intercepted in African plant materials (Appendix A).

### 3.2. Molecular Identification

*Glyphodes pseudocaesalis*. Blasting search against genetic databases highlighted *G. pseudocaesalis* sequence match 98.35% and 99.85% with sequences of the same species available in BOLD and GenBank (accession number AB158235), respectively.*Maruca vitrata*. The two obtained COI sequences of *M. vitrata* were found identical to each other and blast results showed 100% matching in both genetic databases, with several homologous sequences obtained mainly from Indian samples.*Leucinodes africensis*. COI sequence obtained from samples detected in *S. aethiopicum* matched completely with *L. africensis* sequences available in genetic databases referring to samples from Sub-Saharan Africa.*Sternochetus frigidus*. The sequence showed 96.45% identity with sequences of BOLD and GenBank submitted under the specific name.*Anastrepha obliqua*. Blasting search against the databases highlighted that the obtained sequences have more than 99% similarity with sequences of the same species present in both databases and originated in Mexico.*Bactrocera dorsalis*. The findings of several *B. dorsalis* larvae in the seized material showed that specimens from Burkina Faso obtained COI sequences identical to each other as well as for oriental fruit flies inside *Momordica* fruits from Bangladesh. By contrast, specimens found inside guava fruits from Bangladesh showed three different mt-haplotypes. Among them, the mt-haplotype shared by two specimens resulted as being identical both to *B. papayae* and *B. dorsalis* sequences available in the GenBank database. However, ITS1 sequences (accession numbers OL697407-OL697408) confirmed that the collected samples belong to the *B. dorsalis* species. To sum up, no shared mt-haplotypes were found among the Bangladeshi and Burkinabè *B. dorsalis* found in different fruit species seized at Campanian BCPs.

### 3.3. The Borderline Case: An Ecosystem Container

Another interception during the considered period that is noteworthy is what was found during the *Triplochiton scleroxyon* inspection.

The inspection involved some containers containing trunks of *T. scleroxyon*, imported for trade from Cameroon, resulted particularly hardworking and interesting. The inspection covered only the part near the container door because the size and weight of the trunks did not allow an accurate inspection. Inside the containers, several living organisms belonging to different categories of arthropods were found, and the quantity and diversity of the organisms were found to have almost recreated the original ecosystem, with the presence of numerous living insects, spiders, and myriapods. During the survey, a live scorpion (Arachnida, Scorpions) was also found. It was impossible to determine its species because its location in the container prevented its capture. Additionally, it was possible to notice the presence of diffused frass on the trunk examined and on the container bottom, which was produced by the activity of xylophagous organisms. In addition to the frass, there were many holes in the trunks. Thanks to the use of shears and other tools, it was possible to collect samples for subsequent laboratory analysis. Different species of Coleoptera were recognized (Table 1). In particular, *Xyleborus volvulus* was the principal species responsible for the production of the frass.

Although many open tubes of Phostoxin (aluminum phosphide) tablets were found inside the container, this treatment was clearly not sufficient to kill the living arthropods. None of the species found was a quarantine species; however, they are not present in Italy and therefore all of them represent phytosanitary risks. All the containers were rejected and then returned to the sender.

## 4. Discussion

### 4.1. Results of the Inspection Activities of the CNR-IPSP Staff

The last line of defense against the accidental introduction of non-native organisms is represented by phytosanitary inspections at the border control points (BCPs) at ports and airports, where imported goods and passenger baggage are examined [15,33,69,70]. However, control activities at border control points prove more and more every day how global trade and travel represent the main pathways for the introduction of exotic species outside their natural or current ranges and that the number of invasive insects detected globally is increasing [15,71,72].

Although the data reported refer to only a small proportion of Italian points of entry (about 9%), the results in the present work highlight the recurrent and conspicuous entry of exotic species into Italian territory. Unsurprisingly, contextually, there are more and more recent studies that highlight the high number of invasive species found and found to be widespread in Italy in recent decades [19,28,29,67,73,74,75,76]. However, to our knowledge, we cannot know what proportion of the non-indigenous species entering Italy was detected during the same period.

Taking into account only comparable years (2017–2021) (Table 1), Campanian airport and port interceptions were similar (23.9% and 22.7%, respectively), and pests were recovered from exotic fruits brought by travelers who disembarked mainly from Asian countries. Species belonging to Hemiptera and Coleoptera were the most frequently intercepted (Figure 4), showing a complete congruence with the data recorded worldwide [77]; however, the species found are not included in the EPPO quarantine species list.

In imported fruits and vegetables, the main harmful organisms found in this study and in the annual interceptions registered by EUROPHYT (available for years 2016, 2017, and 2018) [17] belong to the fruit flies group, even if the results reported here highlighted that fruit flies are only the fourth most frequent group of insects based on the frequency of detection (Figure 4). However, overall, fruit flies are not the only group of great concern among the intercepted organisms (Table 2) as five additional findings were the most alarming:
(a)In many of the fruits of *Psidium guajava*, *Momordica charantia*, and *Mangifera indica* transported by airline passengers from Bangladesh and Burkina Faso, larval stages of the oriental fruit fly were found (Table 2). This species belongs to the *Bactrocera dorsalis* complex and is considered one of the most harmful pests, recorded in both EPPO A1 [30] and the priority lists of relevant quarantine pests for the EU [31]. Adults are characterized by high mobility, dispersion, fecundity, and, in some species, extreme polyphagy (over 400 host species, many of which are of agricultural interest) [78,79]. The main economic damage associated with this complex is directly linked to the damage to the fruits and the loss of material for exportation due to quarantine restrictions. Results shown here are congruent with data recorded by other authors, according to which specimens of *B. dorsalis* complex are frequently transported by travelers and often intercepted by the plant quarantine service [80]. The oriental fruit fly probably represents the most alarming finding because this species has a high probability of acclimatizing [81,82]. The finding of this species inside the fruits carried by passenger luggage coming predominantly from Bangladesh, the capture of some specimens in the territory of Palma Campania, and the presence of numerous fruit and vegetable food shops owned by Bangladeshis in the same locality suggest the hypothesis that the first field records in Europe [28,29] could be directly correlated with introduction through this route. However, the molecular analyses revealed that none of the specimens found at the BCPs showed an identical mt-haplotype to that found previously in individuals of *B. dorsalis* captured in Italian orchards, for which there is no definitive evidence to support this hypothesis [28,29].(b)Several larvae of *Anastrepha obliqua*, the West Indian fruit fly, were found in some fruits of *Mangifera indica* transported by airline passengers coming from El Salvador. This pest is widespread in some countries of Central and South America [83] and its major host is *M. indica*. This pest is considered a serious threat to all mango-producing regions, and it is included in the EPPO A1 list. Although it was intercepted in the Netherlands in 2013 and in France on mangoes from Mexico [83,84], it has never been intercepted before in Italy. In the event of incursions of this pest, it is critical to focus attention on other hosts that are usually cultivated in the Mediterranean Basin, such as the species of the genus *Citrus*.(c)Larvae of *Leucinodes africensis* were found in *Solanum aethiopicum* fruits transported by airline passengers from Ghana and Bangladesh. This insect is included in the EPPO A1 list and represents a serious phytosanitary threat to Solanaceae crops since this moth damages *Solanum lycopersicum* L. and *Solanum melongena* L. species [58], which are widely cultivated in open fields and in greenhouses in Italy and in the Mediterranean Basin. Recent taxonomical studies [58] highlighted that in Africa several *Leucinodes* species are present, but *Leucinodes orbonalis* Guenée is not. Hence, the 120 *L. orbonalis* intercepted in EPPO countries in the period 2004–2007 in plant material imported from Africa [85,86] should be reconsidered as *L. africensis*. This moth was intercepted twice about a year apart.(d)Adults of *Sternochetus frigidus*, also known as the mango fruit weevil (MFW), were found in fruits of *M. indica* from Burkina Faso. Although this species is native to South East Asia, finding it in fruits from an African country could suggest the spreading of MFW from African fields or the importation of infested mangoes from Asia that are not yet known. MFW is an important economic and quarantine pest for mango [87], but it is actually absent in the EPPO regions. However, recently, mango orchards have moved outside the traditional geographical range, particularly in the Mediterranean area thanks to the suitable sub-tropical climate conditions [88]. Following the spread of this crop in Italy also, mainly in Sicily, the mango is cultivated as a replacement for many crops in which the production and market have suffered losses or crises [89,90]. For this reason, the accidental introduction of the MFW could represent a phytosanitary risk jeopardizing this new agricultural strategy.(e)*Glyphodes pseudocaesalis* larvae were found in *Artocarpus heterophyllus* fruits transported from Bangladesh. This species can pose a serious risk to agriculture because *Glyphodes* spp. belong to the Spilomelini tribe and, in tropical and subtropical regions, these moths are considered the major pests for several economic crops, including citrus, peach, and eggplant [91], which are widely cultivated in the Mediterranean Basin.

### 4.2. Statements about the Inspections

There are several different fundamental variables to be evaluated to reduce the threat linked to the introduction of invasive species: the probability that a harmful organism can enter Mediterranean countries (pathway analysis); the probability that a given plant harbors one type of parasite rather than another; and the possibility that a given parasite can find favorable hosts and climatic conditions and can thereby acclimatize to the newly invaded area (pest risk assessments) [92]. Potential damage from invasive species can be avoided if the invasion is prevented by early detection and intervention, the cheapest approach to managing invasive species [93]. For this aim, a proper quantitative risk assessment is important to better use resources and enhance sampling activities [3].

To comparatively assess a pest evaluation in connection with the risk of introduction, it is essential to take into account the plant species that enter a given region and make an estimation of the probability of entry through a BCP and of the pest acclimatizing to the invaded country. If the unique host plant of a given species is not widespread in the invaded territory, the pest should be considered a minor threat. In contrast, it is self-evident that a polyphagous pest could be a much greater threat as it is able to infest several host plant species because its fitness does not show particular changes when a host shift phenomenon occurs, especially with congeneric hosts [94], and the possibility of acclimatization increases, with the consequent high costs of management [7]. Introduced pests are usually very harmful due to both polyphagy and the absence of natural enemies that could curtail their activity [95,96].

The peculiarity of the “ecosystem container” lies not only in the interception of a high number of still-live species of allochthonous arthropods but also, above all, in the control itself, which occurred under completely exceptional circumstances. That material, in fact, is not regulated, and the control was carried out thanks to the scrupulousness of the forest Carabinieri, who, alarmed by the many insects, alerted the phytosanitary service and support staff.

### 4.3. Claims about the EU Regulations

All the plant material intended for trade within the EU is regulated by the actual EU Plant Health Regulations (see § 1. Introduction) that include controls and inspections of products at the place of production, official registration of the producer, and plant passports issued to accompany the plants and to certify the absence of harmful organisms [34] and generally for the movement of specified plant material within the EU [97]. By contrast, the plant material from third countries is sometimes not adequately inspected because of the lack of equipment, expertise, and infrastructure [45,98].

Passengers often lack such phytosanitary documents due to poor information in this regard from their country of origin. Consequently, the results reported here show that the major threats to South European agro-ecosystems seem to come mainly from the baggage of passengers who are usually unaware that carrying variable quantities of fruits and vegetables can be a threat for introducing new organisms into the incoming countries [99].

The similar incidence of positive cases among plant materials from ports and the airport (~22%, Table 1) indicates that the same level of accuracy applied to controlling the port material should be applied to the airport material too. This was especially true because the incoming airport material, compared to that from the port, showed a greater diversity of plant species although in limited amounts. However, the interceptions shown here, and in any case also those recorded in the phytosanitary databases, could represent only the tip of the iceberg for a much bigger problem. On the one hand, in fact, the passengers’ material is not controlled and/or blocked in its entirety, and on the other hand, a good part of the material arriving at the port is not controlled as it is excluded from regulated materials.

The expansion of international trade in commodities constrains the ability of phytosanitary inspectors by allowing them to sample only a small part of the total imported goods [45,100]. However, this work has proven that goods carried by airline passengers represent a risk of the introduction of new alien pests that cannot be underestimated. The import of plants and plant products through this means is not yet quantified, and deep risk analysis is needed. In the USA, for example, it has been estimated that about 2% of all border crossing cargo is inspected, including inspections at maritime ports, airports, and land crossings [101]. The high costs of management for new introductions of harmful organisms in terms of direct and indirect damage and in terms of controls, eradication, and containment measures highlight the strategic role of the phytosanitary service in conjunction with international institutions for plant protection [102]. Indeed, it is widely known that costs associated with post-invasion management and damage are much higher than those associated with pre-invasion management [102,103].

Due to the impossibility of conducting an integral check of all of the material transported in Europe, we propose the following possible solutions: (1) including a highly specialized professional at passenger baggage checks who will pay particular attention during detection, in particular, of illegal imports [98]; (2) optimizing and increasing checks on passenger baggage in relation to point of origin [15]; (3) employing diagnostic tests during checks for the early detection of harmful organisms [104] by exploiting available technologies and application methodologies (X-ray scanning techniques, sniffer dogs, and human inspection) [99].

Additionally, it is essential to improve communication about plant health risks between the various actors involved in the trade of plant material and between those who intend to transport it in their luggage. For example, creating infographics and factsheets would be useful for correctly informing passengers and would be crucial in making them responsible for and conscious of the risks linked to the transport of plant material and their pests from their country of origin. EPPO already started an important advertising campaign a few years ago to raise public awareness and to encourage responsible behavior [105]. This campaign should progress during the years and should be updated with the new threats. If the advertising campaign has no satisfactory effects, the need to establish a strong deterrent is mandatory. It could discourage the transport of this infested material from the country of origin through a system of fines for non-compliant plant material found inside passenger luggage.

Finally, creating a chronological database of the interceptions at BCPs can be useful for understanding how organisms can enter a region, which are the pathways, with what frequency they enter, and whether there is a correlation between an intercepted species and its presence in a given territory. Additionally, sharing the results of the interceptions, in terms of organisms and of means of introduction, could be an important benefit [77].

## 5. Conclusions

The data reported in this study suggest that customs controls and, in particular, the controls of the material imported by passengers in their luggage represent a very high risk for the possible introduction of invasive alien species. In fact, the interceptions carried out in a couple of BCPs in the same region, with regular checks at the port BCPs and irregular and random checks at the airport ones, highlight several interceptions of threatening pests, and often the detected species were very dangerous invasive alien species.

Therefore, investing in the number of checks at BCPs is crucial, but the relevance of the efficiency of the inspections of the plant materials should not be underestimated. The financial expense of the pre-invasion management (such as the training and hiring of highly specialized personnel, laboratory analyses, and the ancillary costs related to the materials’ transport and storage) are far lower than the costs for the management of the agronomic and environmental impact that incursions of one or more harmful species could cause.

## Figures and Tables

**Figure 1 insects-13-00617-f001:**
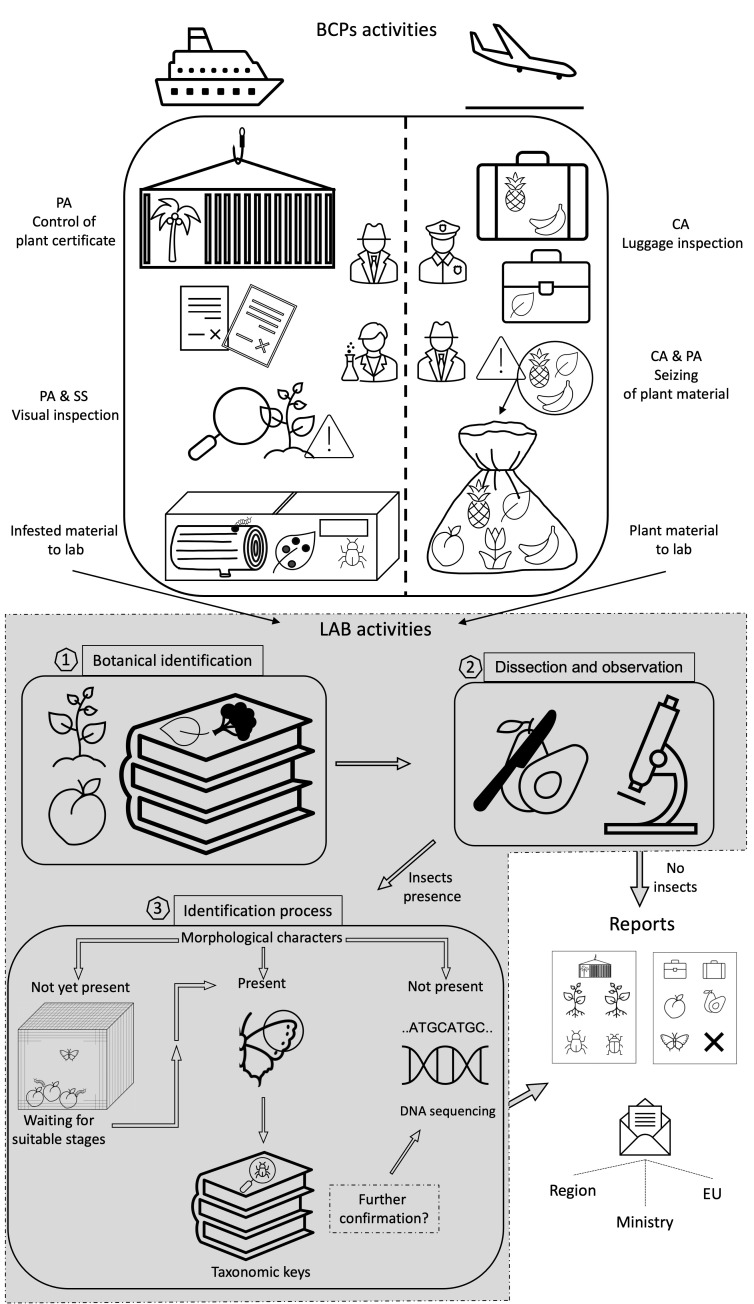
Summary workflow for BCPs and lab activities aimed at intercepting and identifying infested plant materials and pests. PA, phytosanitary agent; CA, customs authority; SS, scientific support.

**Figure 2 insects-13-00617-f002:**
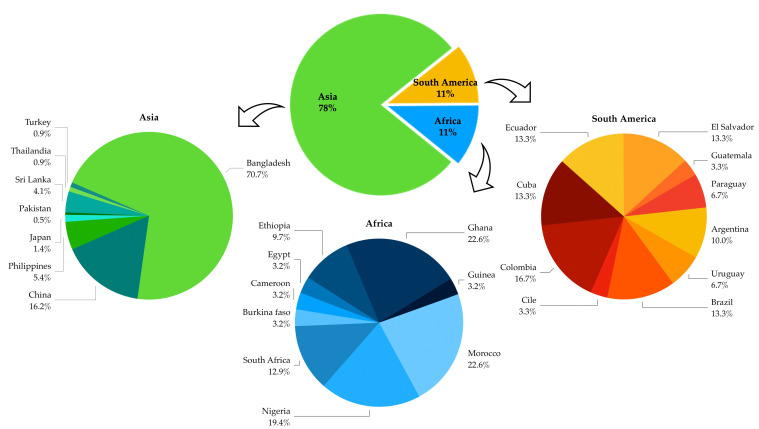
Country of origin of the plant material seized at the BIPs, (Turkey was included in Asian countries).

**Figure 3 insects-13-00617-f003:**
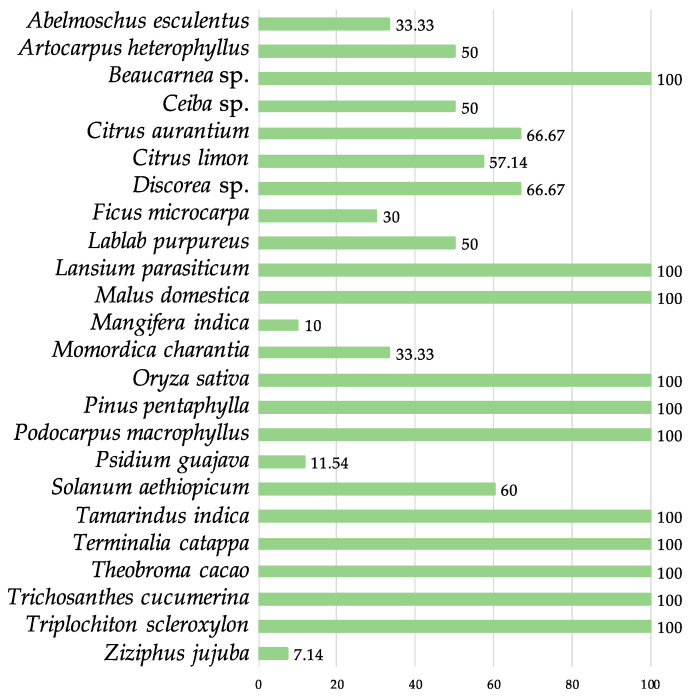
Percentage of infestation of the plant material for the different species examined.

**Figure 4 insects-13-00617-f004:**
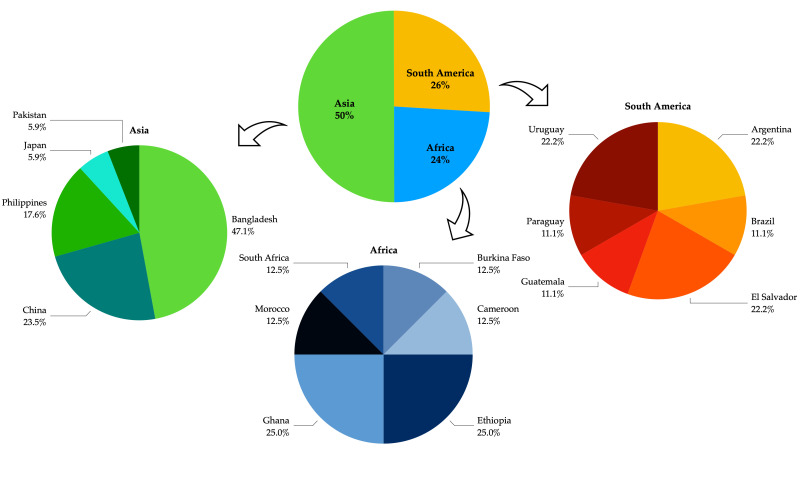
Percentage of infested plant material seized at BIPs based on country of origin.

**Figure 5 insects-13-00617-f005:**
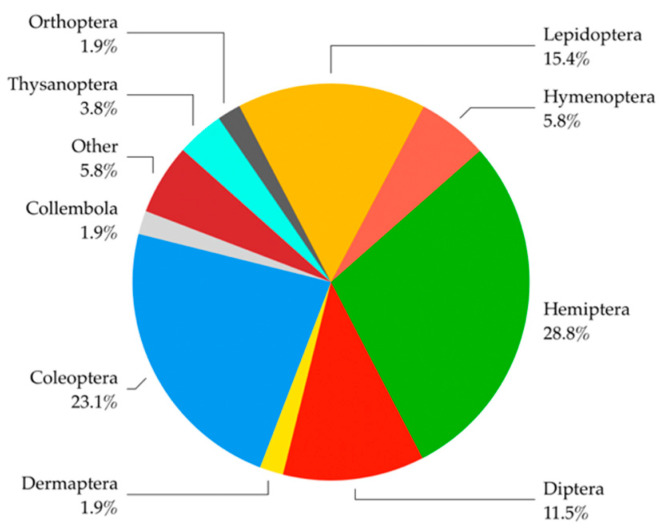
Relative frequency of orders to which the detected pests belong (Collembola were included due to their possible phytosanitary risk).

**Table 1 insects-13-00617-t001:** Number of inspections from 2016 to 2021 and respective number of detected pests in fruits, seeds, parts of plant, wood, and other plant materials. * Data refer to the first half of 2021.

Year	Port		Airport		Plant Species	Pest Species Detected
	Inspection	Positive	Inspection	Positive		
2016	19	10	-	-	17	16
2017	7	2	3	2	21	4
2018	5	1	17	2	45	4
2019	2	0	17	3	35	9
2020	1	0	7	3	34	6
2021 *	7	2	2	1	16	6
TOT	41	15	46	11	168	45

**Table 2 insects-13-00617-t002:** Main pests and relative hosts detected during inspections at BCPs of Campania Region. u, unit; AP, International Airport of Naples; PN, Port of Naples; PS, Port of Salerno; Mpl, morphology; Mol, molecular. Col, Coleoptera; Hem, Hemiptera; Lep, Lepidoptera; Thy, Thysanoptera; Hym, Hymenoptera; Dip, Diptera; Der, Dermaptera; Dipl, Diplopoda; lv, larvae; †, dead specimen.

Host Botanical Name	Year	Quantity	Part of Plant	Country of Origin	BCP	Methods	Pest
*Abelmoschus esculentus* (L.) Mönch	2020	8 kg	fruits	Bangladesh	AP	Mpl	Lep: Nolidae: *Earias vittella* (Fabricius)
*Artocarpus heterophyllus* Lam.	2020	1 kg	fruits	Bangladesh	AP	Mpl/Mol	Lep: Crambidae: *Glyphodes pseudocaesalis* Kenrick lv. †
*Beaucarnea* sp. Lam.	2017	20,000 u	plants	Guatemala	PN	Mpl	Col: Dryophthoridae: *Scyphophorus acupunctatus* (Gyllenhal)
*Ceiba* sp. Mill.	2018	4 u	trunks (~12 m)	Paraguay	PN	Mpl	Col: Cerambycidae: *Steirastroma breve* (Sulzer)
*Citrus limon* (L.) Burm. F.	2016	40 kg	fruits	Argentina	AP	Mpl	Hem: Diaspididae: *Aspidiotus nerii* (Bouché)
	2016	30 kg	fruits	Argentina	PN	Mpl	
*Citrus X aurantium* L.	2016	170 kg	fruits	Uruguay	PN	Mpl	Hem: Diaspididae: *Aspidiotus* sp., *Lepidosaphes beckii* (Newman)
	2016	40 kg	fruits	South Africa	PN	Mpl	Hem: Diaspididae: *Aspidiotus* sp.
*Dioscorea* sp. L.	2021	12,000 kg	tubers	Ghana	PS	Mpl	Hem: Pseudococcidae: *Planococcus citri* (Risso); Col: Dermestidae: *Dermestes maculatus* (De Geer)
*Ficus microcarpa* L. f.	2016	1200 u	plants	China	PN	Mpl	Hym: Agaonidae: *Josephiella microcarpae* (Beardsley and Rasplus); Thy: Phlaeothripidae: *Gynaikothrips ficorum*; Hem: Coccidae: *Saissetia oleae* (Olivier)
	2019	200 u	plants	China	PS	Mpl	Hem: Coccidae: *Ceroplastes floridensis* (Comstock), *Lecanium* sp.
	2021	50 u	plants	China	PN	Mpl	Thy: Phlaeothripidae: *Gynaikothrips ficorum*; Der: Forficulidae: *Forficula* sp. (Linnaeus); Dipl: Polydesmida: Polydesmidae: *Polidesmus* sp. (Pocock)
*Lablab purpureus* (L.) Sweet	2021	0.7 kg	pods	Bangladesh	AP	Mpl/Mol	Lep: Crambidae: *Maruca vitrata* Fabricius 2 lv. †
*Lansium domesticum* Corrêa	2018	3 kg	fruits	Philippines	AP	Mpl	Hem: Pseudococcidae: *Phenacoccus aceris* (Signoret)
*Malus domestica* Borkh.	2018	2 kg	fruits	Morocco	AP	Mpl	Lep: Tortricidae: *Cydia pomonella* (Linnaeus)
*Mangifera indica* L.	2019	15 kg	fruits	Burkina Faso	AP	Mpl/Mol Mol	Col: Curculionidae: *Sternochetus frigidus* (Fabricius); Dip: Tephritidae: *Bactrocera dorsalis* (Hendel) 2 lv. †
	2020	2 kg	fruits	El Salvador	AP	Mol	Dip: Tephritidae: *Anastrepha obliqua* Macquart lv. †
*Momordica charantia* L.	2020	5 kg	fruits	Bangladesh	AP	Mpl/Mol	Dip: Tephritidae: *B. dorsalis* 2 lv. †
*Oryza sativa* L.	2017	2 kg	caryopsis	Pakistan	PN	Mpl	Col: Cucujidae: *Cryptolestes ferrugineus* (Stephens)
*Pinus parviflora* Siebold & Zucc.	2016	4 u	plants	China	PN	Mpl	Hem: Adelgidae: *Pineus* sp.
*Podocarpus macrophyllus* (Thunb.) Sweet	2016	24 u	trunks	Japan	PN	Mpl	Hem: Coccidae: *Ceroplastes ceriferus* (Fabricius)
*Psidium guajava* L.	2018	1 u	fruits	Brazil	AP	Mpl	Hem: Pseudococcidae: *Planococcus citri* (Risso)
	2019	56 kg	fruits	Bangladesh	AP	Mpl/Mol	Hem: Pseudococcidae: *P. citri*; Dip: Tephritidae: 3 adults and 1 lv. of *Bactrocera dorsalis* complex
*Solanum aethiopicum* L.	2017	2 kg	fruits	Ghana	AP	Mpl	Lep: Crambidae: *Leucinodes* sp.
	2017	2 kg	fruits	Ethiopia	AP	Mpl	Lep: Crambidae: *Leucinodes africensis* Mally, Korycinska, Agassiz, Hall, Hodgetts and Nuss
	2019	10 kg	fruits	Bangladesh	AP	Mpl/Mol	Lep: Crambidae: *L. africensis*
*Tamarindus indica* L.	2019	5 kg	fruits	Philippines	AP	Mpl	Col: Bruchidae: *Caryedon serratus* (Oliver); Dryophthoridae: *Sitophilus linearis* (Herbst)
*Terminalia catappa* L.	2020	50 u	leaves	El Salvador	AP	Mpl	Hym: Formicidae: *Camponotus* sp.
*Theobroma cacao* L.	2017	5 kg	fruits	Ethiopia	AP	Mpl	Diplopoda
*Triplochiton scleroxylon* K. Schum.	2016	16 u	trunks	Cameroon	PN	Mpl	Col: Curculionidae: *Doliopygus* sp. (Schedl), *Xyleborus volvulus* (Fabricius); Cerambicidae: *Ancylonotus tribulus* (Fabricius); Histeridae: *Hololepta plana*; Col: Carabidea; Hem: Miridae; Hym: Formicidae; Orthoptera; Aracnida: Scorpiones
*Ziziphus jujuba* Mill.	2019	23 kg	fruits	Bangladesh	AP	Mpl	Lep: Pyralidae: species near *Sciota subcaesiella* Clemens and *Meyrickiella homosema*

## Data Availability

Data is contained within the article or Appendix A. Raw data are available on request from the corresponding author; DNA sequences were submitted to the publicly accessible database GenBank (https://www.ncbi.nlm.nih.gov/genbank/).

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
