# Peer review of "The Bugs in the Bags: The Risk Associated with the Introduction of Small Quantities of Fruit and Plants by Airline Passengers"

_insects, 2022, doi:10.3390/insects13070617_

Round 1

Reviewer 1 Report

This is a very important paper reporting the interception records of exotic insects in the ports and airports of Campania, Italy. The manuscript is well written and almost acceptable. The following are some minor suggestions you might want to consider.

Line 18 'figure': The meaning of the 'figure' here is difficult to imagine. After reading the discussion I understood it means 'person' but perhaps you had better reword 'figure' to some other appropriate word.

Introduction, line 97-133: Perhaps the description is too detailed and away from science. I understand the regulation is important but it might be less important to non-EU readers and turn off some of them. Indeed, I was able to understand the rest of the manuscript without reading these details. I would like to leave it up to the authors, but please consider moving the details to appendix or supplementary material.

Line 169: Please clarify whether the inspection was carried out for all of the plant materials imported. After reading line 458, I understand that only a part of them was inspected, but this should be clarified at the beginning.

Line 182 'seized by the custom authority': What was the criteria for inspection?

Line 400 'mango seed weevil': MFW?

Line 469: Again, I am not familiar with the use of 'figure' here.

That's all. Thank you.

Author Response

Response to Reviewer 1 Comments

Thank you very much for your suggestions and advices, our responses point by point are in red.

Point 1) Line 18 'figure': The meaning of the 'figure' here is difficult to imagine. After reading the discussion I understood it means 'person' but perhaps you had better reword 'figure' to some other appropriate word.

 Response 1) Done. We changed the word to “position”.

 Point 2) Introduction, line 97-133: Perhaps the description is too detailed and away from science. I understand the regulation is important but it might be less important to non-EU readers and turn off some of them. Indeed, I was able to understand the rest of the manuscript without reading these details. I would like to leave it up to the authors, but please consider moving the details to appendix or supplementary material.

 Response 2) Done. We moved a large part of the regulation in the supplementary material.

Point 3) Line 169: Please clarify whether the inspection was carried out for all of the plant materials imported. After reading line 458, I understand that only a part of them was inspected, but this should be clarified at the beginning.

Response 3) Done, we clarified our statement by adding “to regulated material” (line 173).

Point 4) Line 182 'seized by the custom authority': What was the criteria for inspection?

Response 4) Done, we added the reason based on the controls were carried out: “…because without phytosanitary certificate” (line 194).

Point 5) Line 400 'mango seed weevil': MFW?

Response 5) Done, it was a typo in the previous version of the manuscript. We corrected the acronyms in MFS. (line 422)

Point 6) Line 469: Again, I am not familiar with the use of 'figure' here.

Response 6) Done. We changed the word to “position”. (line 493)

Best regards

Francesco Nugnes on behalf of all co-authors

Reviewer 2 Report

This manuscript emphasized the importance of checking the plant material that can be imported into the baggage of airline passengers where invasive pests can come through, the work is also part of the quarantine task worldwide, main comments were as follows:

It`s better to draw a workflow showing the procedures and main techniques through this part of quarantine measures and also compare with the other European countries and worldwide.

Proportion of the origin countries and plant materials were analyzed from the intercepted data, the correlation with the trade and other factors was suggested to be analyzed.

Line 294,  B. papayae was proved to be the same species as B. dorsalis in 2014. Bactrocera dorsalis complex was not only referred to the oriental fruit fly but also more than 70 species.

Line 367-370, the conclusion of the origin of B. dorsalis should be cautiously, haplotype information from only two mitochondrial sequences could not draw this conclusion.

Author Response

Response to Reviewer 2 Comments

Thank you very much for your suggestions and advices, our responses point by point are in red.

Point 1) It`s better to draw a workflow showing the procedures and main techniques through this part of quarantine measures and also compare with the other European countries and worldwide.

Point 2) Proportion of the origin countries and plant materials were analyzed from the intercepted data, the correlation with the trade and other factors was suggested to be analyzed.

Response 1 and 2) The manuscript here presented refers to our supporting activities to Phytosanitary agents at BCPs of the Campania region, some factors were not available (especially the not regulated materials or the plant materials in passenger luggage). Our work lays the foundations for a bigger project with the participation of other researchers (from Italy or other countries) involved in similar activities at BCPs where other factors will be correlated, and the procedures of quarantine measures will be compared. However, we accepted the referee’s suggestion to supply the manuscript with a workflow (figure 1).

Point 3) Line 294,  B. papayae was proved to be the same species as B. dorsalis in 2014. Bactrocera dorsalis complex was not only referred to the oriental fruit fly but also more than 70 species. 

Response 3) We agree with the referee’s statement. However, in the “results” paragraph we showed only the identity rates obtained from GenBank blasting with available sequences (lines 314-316).

Point 4) Line 367-370, the conclusion of the origin of B. dorsalis should be cautiously, haplotype information from only two mitochondrial sequences could not draw this conclusion.

Response 4) The paragraph (from lines 383 to 392) should be taken in its entirety, and our opinion is very cautious. Based on our findings and our analyses, we only speculated about the origin of B. dorsalis, but we did not state with certainty the route of invasion or the origin of the recorded Oriental fruit flies.

Best regards

Francesco Nugnes on behalf of all co-authors

Reviewer 3 Report

1. The introduction is very large. It should be slightly shortened. It is also necessary to clearly state the goals and objectives of the study.

2. Line 180-184. Why did passengers carry these products and raw materials? Why was it seized from them at customs?

3. List the guides for plant identification.

4. Why are more plant materials imported from Bangladesh? Is it possible that the ports and the airport are the main hub for flights from this country?

5. Line 450-451. Why is there a difference in the material from the ports and the airport?

6. Line 470-474. What financial expenses will there be if all your recommendations are taken into account? Indicate this in the conclusion.

7. There is no conclusion. The authors should do it.

Author Response

Response to Reviewer 3 Comments

Thank you very much for your suggestions and advices, our responses point by point are in red.

Point 1) The introduction is very large. It should be slightly shortened. It is also necessary to clearly state the goals and objectives of the study.

Response 1) Done. Following your and the other referee’s advices, we shortened the introduction, moving the regulation paragraph to supplementary materials and adding some clarification about the study’s aims.

Point 2) Line 180-184. Why did passengers carry these products and raw materials? Why was it seized from them at customs?

Response 2) Passengers carried these products mainly for personal consumption and sometimes local market (we added this statement at line193). We added the seizing reason in line 194 “…because without phytosanitary certificate”.

Point 3) List the guides for plant identification.

Response 3) Done, we added information in lines 199-200 and respective bibliography.

Point 4) Why are more plant materials imported from Bangladesh? Is it possible that the ports and the airport are the main hub for flights from this country?

Response 4) In the paragraph about Bactrocera dorsalis findings (section 4.1 letter a) we already stated that “the presence of numerous fruit and vegetable food shops owned by Bangladeshis in the same locality”. In some localities of the Campania region, there is a large presence of foreign populations, in particular Asian ones. Therefore, it is conceivable the existence of the transport of products for simple personal consumption or for sale.

Point 5) Line 450-451. Why is there a difference in the material from the ports and the airport?

Response 5) There was no significant difference in the number of positive detections recorded by port and airport, but there was a difference in the number of plant species transported, with greater diversity in species arriving at the airport. We stressed the greater diversity of plant species found at the airport, although in limited amounts.

Point 6) Line 470-474. What financial expenses will there be if all your recommendations are taken into account? Indicate this in the conclusion.

Point 7) There is no conclusion. The authors should do it.

Response 6 and 7) Lines 489-490 and 519-528. We added some information about the costs and respective bibliography in the discussion paragraph. Furthermore, a conclusion was added, where we dealt with the economic topic considering the different financial expenses in pre-invasion management such as the training of highly specialized personnel and its hiring, analyses, transport, and storage costs.

Best regards

Francesco Nugnes on behalf of all co-authors